

# The higher relative concentration of K⁺ to Na⁺ in saline water improves soil hydraulic conductivity, salt leaching efficiency and structural stability

Sihui Yan [a, b], Binbin Zhang [a, b], Tonggang Zhang [a, b], Yu Cheng [a, b], Chun Wang [a, b], Min
Luo [a, b], Hao Feng [a, c], Tibin Zhang [a, c, *], Kadambot H.M. Siddique [d]
[a.] Key Laboratory of Agricultural Soil and Water Engineering in Arid and Semiarid
Areas, Ministry of Education, Northwest A&F University, Yangling, Shaanxi 712100,
P. R. China
[b.] College of Water Resources and Architecture Engineering, Northwest A&F
University, Yangling, Shaanxi 712100, P. R. China
[c.] Institute of Soil and Water Conservation, Northwest A&F University, Yangling,
Shaanxi 712100, P. R. China
[d] The UWA Institute of Agriculture, The University of Western Australia, Perth, WA
6001, Australia
* Corresponding author at: Institute of Soil and Water Conservation, Northwest A&F
University, Yangling, Shaanxi 712100, China. Tel.: +86 29 87012871; fax: +86 29
87011354. E-mail: zhangtibin@163.com (T. Zhang)



**Abstract**
Soil salinity and sodicity caused by saline water irrigation are widely observed
globally. Clay dispersion and swelling are influenced by sodium ($Na^+$) concentration
and electrical conductivity (EC) of soil solution. Specifically, soil potassium ($K^+$) also
significantly affects soil structural stability, but which concern was rarely addressed in
previous studies or irrigation practices. A soil column experiment was carried out to
examine the effects of saline water with different relative concentrations of $K^+$ to $Na^+$,
including $K^+/Na^+$ of 0:1 (K0Na1), 1:1 (K1Na1), 1:0 (K1Na0) at a constant EC (4 dS m$^{-1}$
$^1$), and deionized water as the control (CK), on soil physicochemical properties. The
results indicated that at the constant EC of 4 dS m$^{-1}$, the infiltration rate and water
content were significantly ($P < 0.05$) affected by $K^+/Na^+$ values, K0Na1, K1Na1 and
K1Na0 significantly ($P < 0.05$) reduced saturated hydraulic conductivity by 43.62%,
29.04% and 18.06% respectively compared with CK. The volumetric water content was
significantly ($P < 0.05$) higher in K0Na1 than CK at both 15 and 30 cm soil depths.
K1Na1 and K1Na0 significantly ($P < 0.05$) reduced the desalination time and required
leaching volume. K0Na1 and K1Na1 reached the desalination standard after the fifth
and second infiltration, respectively, as K1Na0 did not exceed the bulk electrical
conductivity required for desalination prerequisite throughout the whole infiltration
cycle at 15 cm soil layer. Furthermore, due to the transformation of macropores into
micropores spurred by clay dispersion, soil total porosity in K0Na1 dramatically
decreased compared with CK, and K1Na0 even increased the proportion of soil





macropores. The higher relative concentration of $K^+$ to $Na^+$ in applied water was more
conducive to soil aggregates stability, alleviating the risk of macropores reduction
caused by sodicity.
**Keywords:** Saline water; Cation composition; Hydraulic properties; Desalination; Pore
structure.
**1 Introduction**

Freshwater shortage resulted from elevated demand for water resources as well as

the irrational exploitation and use after economic and population growth (Zhang and
Xie 2019; Prajapati et al. 2021), constrains the sustainability of agricultural production
(Aparicio et al., 2019). Alternative water resources with variable water quality (such as
saline water) are being considered for agricultural irrigation in several desert and saline
areas (Singh et al. 2021; Liu et al. 2022a). Utilizing saline water could partly alleviate
the undersupply of freshwater for agricultural production (Yang et al., 2020). However,
the other side of the coin is that saline water irrigation could result in soil salinization
and/or sodicity. This disaster is related directly to soil pore size distribution, and in turn
to the dispersion and swelling of the clay fraction (Bouksila et al. 2013; Hack-ten
Broeke et al. 2016; Zhang et al. 2018). Therefore, in order to optimize saline water
utilization, the effects of saline water quality on the soil hydraulic properties and pore
structure characteristics should be paid more attention (Scudiero et al., 2017).

Saline water irrigation can increase the monovalent ions concentration in soil



solution and affect soil structure (Qadir et al. 2007; Qadir et al. 2021). Excess sodium
(Na⁺) from irrigation saline water is adsorbed onto the clay exchange surface in salt-
affected soils where sodium compounds predominate contributing to the disintegration
of soil structure (Marchuk and Rengasamy 2011; Belkheiri and Mulas 2013; Awedat et
al. 2021). As percolation progresses, the thickness of the diffusion double electron
layers raises due to the relatively larger hydrated radius of Na⁺, and the repulsive force
between adjacent diffusion double electron layers appears to increase, resulting in the
dispersion and swelling of soil particles (Alva et al. 1991; Reading et al. 2015).

Soil calcium ($Ca^{2+}$) and magnesium ($Mg^{2+}$) can alleviate soil dispersibility by

replacing Na⁺ in soil colloids, the outer layer of the $Ca^{2+}$ and $Mg^{2+}$ containing colloidal
particles do not adsorb water molecules, turning Na⁺ qualitative hydrophilic colloid into
$Ca^{2+}$ and $Mg^{2+}$ hydrophobic colloid (Marchuk and Rengasamy 2011; Tsai et al. 2012).
Colloidal particles get close to each other, promoting soil particles forming water stable
aggregates, thus improving soil structural stability (Gharaibeh et al. 2009; McKenna et
al. 2019). Therefore, the concentration of Na⁺ in relation to $Mg^{2+}$ and $Ca^{2+}$ (Sodium
adsorption ratio, SAR) (U.S. Salinity Laboratory Staff 1954) is considered a crucial
criterion in soil structural stability and hydraulic conductivity (Rengasamy and
Marchuk 2011). Although SAR can be used to predict soil clay dispersion effect caused
by cations, the controlling mechanism of dispersion in SAR is presumed to be
exchangeable Na⁺. However, Na⁺ does not alone cause soil dispersion since the
chemical component of clay structure integrity is mainly a function of ionic valence



and hydration radius (Marchuk et al., 2014). Potassium ($K^+$) has been overlooked
because salt-affected soils typically contain low amounts of $K^+$. However, Li et al.
(2022) reported that under the continuous recycling use of underground saline water,
water-soluble and exchangeable $K^+$ is higher than $Ca^{2+}$ and $Mg^{2+}$ in the Hetao irrigation
district—one of the large irrigation districts in China. It is anticipated that the long-term
use of irrigation water with high $K^+$ concentrations may therefore create substantial
challenges in preserving good soil structure and adequate infiltration rates (Sposito et
al., 2016). $K^+$ is not as effective as $Na^+$ in generating soil particle dispersion and
swelling problems, yet Marchuk and Marchuk (2018) pointed out that $K^+$ could
substitute $Na^+$ on exchange sites to encourage $Na^+$ leaching and increase water
conductivity to some extent. A lower concentration of $K^+$ may have positive effects on
soil permeability due to the substitution of exchangeable $Na^+$ by $K^+$ with lower
dispersive potential, increasing aggregates stability and soil pore connectivity (Buelow
et al., 2015).

Thus, we hypothesized that the amount of $K^+$ relative to $Na^+$ would certainly have

an effect on soil structural stability, and these relationships could be well evaluated by
a column experiment under controlled conditions. The specific objectives of this study
were to (1) ascertain the effect of irrigation saline water with different relative
concentrations of $K^+$ to $Na^+$ ($K^+/Na^+$) on transport and distribution of water and salt; (2)
determine the effect on soil pore structural characteristics; (3) predict these effects using
a newly proposed index (CROSS) rather than SAR.





**2 Materials and methods**
**2.1 Soil sampling location and properties**

The study soil was collected from a layer of 0-40 cm field in Yangling (108°04′E,

34°20′N), Shaanxi Province, China. After air-dried, the soil was grounded to pass
through a 2-mm seize. Soil physical and chemical properties are listed in Table 1. Soil
particle size distribution was measured by the Laser Mastersizer 2000 (Malvern
Instruments, Malvern, UK), and according to the USDA classification system, soil
texture is classified as silty clay loam. Soil bulk density calculated by soil core method
was 1.35 g cm$^{-3}$. The soil had a low salt concentration with $EC_e$ (electrical conductivity
of saturated extract) of 0.72 dS m$^{-1}$ and pH of 7.66, respectively. $EC_e$ and pH were
measured by conductivity meter (DDS-307, China) and pH meter (PHS-3C, China),
respectively. Flame photometry (6400A, China) was used to measure soluble Na$^+$ and
K$^+$, $CO_3^{2-}$ and $HCO_3^-$ concentrations were tested by the neutral titration method, Cl$^-$ was
analyzed by the silver nitrate titration method, and $SO_4^{2-}$ was determined by barium
sulfate turbidimetric method, Mg$^{2+}$ and Ca$^{2+}$ were specified using ethylene diamine
tetraacetic acid (EDTA) titrimetric method (Bao 2005).






Table 1 The physicochemical properties of study soil.

| Property | Value |
| --- | --- |
| Particle size distribution (%) | |
| Sand (> 0.05 mm) | 8.10 |
| Silt (0.05-0.002 mm) | 60.62 |
| Clay (<0.002 mm) | 31.28 |
| Texture | Silty clay loam |
| $EC_e$ (dS m$^{-1}$) | 0.72 |
| pH | 7.66 |
| Ion concentration (mmol L$^{-1}$) | |
| $CO_3^{2-}+HCO_3^-$ | 0.60 |
| $Cl^-$ | 0.23 |
| $SO_4^{2-}$ | 2.18 |
| $Mg^{2+}$ | 0.32 |
| $Ca^{2+}$ | 0.54 |
| $Na^+$ | 0.10 |
| $K^+$ | < 0.01 |

Note: $EC_e$ represents electrical conductivity of soil saturated extract.
**2.2 Experiment design**

Soil columns were prepared using transparent polyvinyl chloride cylinders, with

an internal diameter of 20 cm and a height of 50 cm (Fig. 1). Round and small holes (6
mm diameter) were equally arranged at the bottom for drainage. A 5 cm depth quartz
sand was laid at the bottom of the soil column as a filter layer before packing to prevent
small soil particles from being washed away. After that, air-dried soil was packed at 40
cm height with a bulk density of 1.35 g cm$^{-3}$ (referring to the original level of the soil).
The sieved dry soil was poured into each soil column in the 5-cm sections for uniform
compaction, and the layer's surface was roughened to ensure a tight connection to the
next layer. The soil column was then allowed to stand in the laboratory for 24 hours



before the initiation of the experiments described herein. The constant water head (2
cm, using a Mariotte bottle) infiltration experiment was conducted with three
replications for each treatment.

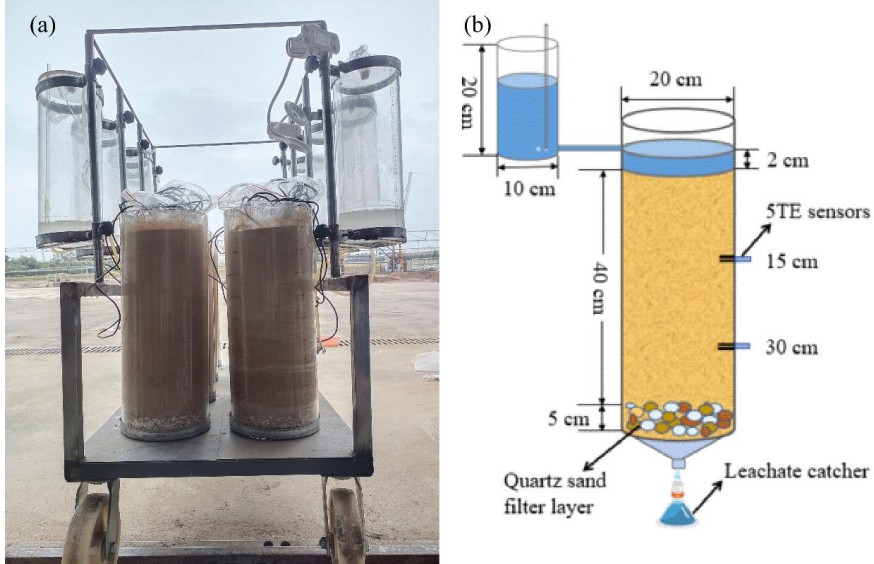


Fig. 1. Illustration of the experiment apparatus (a) and schematic diagram (b).

Three infiltration solutions were prepared with different ratios of $K^+/Na^+$ (0:1

(K0Na1), 1:1 (K1Na1), 1:0 (K1Na0) at constant EC of 4 dS m$^{-1}$ and deionized water as
the control (CK) (Table 2). The cation ratio of soil structural stability (CROSS)
(Rengasamy and Marchuk 2011) was an indicator of soil structural behavior as
influenced by both $Na^+$ and $K^+$, and it was calculated accordingly (Smith et al., 2015):

$$CROSS = \frac{Na^+ + 0.335K^+}{\left[\left(Ca^{2+} + 0.0758Mg^{2+}\right)/2\right]^{0.5}}$$                 (1)

where different chemical element symbols denote charge concentrations (mmol$_c$ L$^{-1}$).



Table 2 Saline water settings with different K⁺/Na⁺ at a constant EC.

| Treatment | Adding salt/ (mmol L$^{-1}$) | | | K$^+$/Na$^+$ | Setting EC (dS m$^{-1}$) | Measured EC (dS m$^{-1}$) | CROSS (mmol$_c$ L$^{-1}$)$^{0.5}$ |
|---|---|---|---|---|---|---|---|
| | KCl | NaCl | CaCl$_2$ | | | | |
| K0Na1 | 0 | 34 | 3 | 0:1 | 4.00 | 4.25 | 27.76 |
| K1Na1 | 17 | 17 | 3 | 1:1 | 4.00 | 4.33 | 17.49 |
| K1Na0 | 34 | 0 | 3 | 1:0 | 4.00 | 4.40 | 7.22 |
| CK | Deionized water | | | / | 0.00 | 0.02 | / |

Note: K0Na1, K1Na1 and K1Na0 indicate the saline water at EC of 4 dS m$^{-1}$ with
K$^+$/Na$^+$ of 0:1, 1:1 and 1:0, respectively; CK, deionized water; CROSS represents cation
ratio of soil structural stability.
The experiment was implemented in the form of alternate leaching, the prolonged
leaching process of soil substrates proved more useful for illuminating the function of
electrolyte effect and cation exchange (Shaygan et al., 2017). The next infiltration was
performed two days after the drainage of the previous infiltration was completed. Soil
layers were regarded as reaching desalination prerequisite when the soil salt content
came to less than 0.3%, which meant that bulk electrical conductivity was less than 1.5
dS m$^{-1}$ (transformation from salt content to bulk electrical conductivity) (Yin et al.,
2022). Water application was stopped when the bulk electrical conductivity of all
treatments at 15 cm depth reached the prerequisite for desalination. This experiment
was planned to fill all the pores in the soil column throughout the infiltration cycle,
therefore the water applied at the first infiltration was described by the pore volume
equation (Xu and Huang 2010):
$$V_p = V_s \cdot TP \qquad (2)$$



$$TP = \frac{ds - BD}{ds}$$
(3)

where $V_p$ is the pore volume (cm$^3$), $V_s$ is the volume of filled soil (cm$^3$), $TP$ is the soil
total porosity (%), $ds$ is the soil particle density (2.65 g cm$^{-3}$) (Xu and Huang 2010),
$BD$ is the bulk density (g cm$^{-3}$). According to Eq. (2) and Eq. (3), around 6 L of water
was required in the first infiltration. Required water volume for each subsequent
leaching was determined by the volume of leachate at the first infiltration, 0.5 L each
time.
**2.3 Measurements**
During the whole experiment period, soil volumetric water content and bulk
electrical conductivity were real-time monitored at 15 and 30 cm soil depths from the
soil surface by capacitance sensors (ECH2O 5TE, METER Group, USA) (Fig. 1). The
leachate was collected in the leachate catcher below the soil column. Cumulative
leachate volume was monitored over time to determine the saturated hydraulic
conductivity ($K_{sat}$, cm min$^{-1}$) of each treatment by using a derivation of Darcy's
approach (Sahin et al., 2011):
$$K_{sat} = \frac{V_l \cdot H}{A \cdot t \cdot (H + h)}$$
(4)

where $V_l$ is the leachate volume (cm$^3$), $H$ is the length of filled soil (cm), $A$ is the surface
area of soil column (cm$^2$), $t$ is the leaching time of measurement (min), $h$ is the height
of constant water head (cm).
The salt accumulated in the soil column was determined by subtracting the salt in



the leachate from the applied water, the salination rate ($Rs$, %) indicated the ratio of salt
accumulated in the soil column at every time of infiltration to the salt content at the first
applied water. Leaching efficiency ($Le$, g L$^{-1}$) referred to the amount of desalination per
unit of water volume in the desalination process. $Rs$ and $Le$ were calculated as follows:

$$Rs = m_s / m_w \tag{5}$$

$$Le = (m_s - m_1) / w \tag{6}$$

Where $m_s$ is the salt content accumulated in soil column at every time of infiltration (g),
$m_w$ is the salt content in the total water used for the first infiltration (g), $m_1$ is the mass
of salts after the first infiltration (g), $w$ is the total water volume used for leaching (L).

Soil samples were collected from each soil column at 5-cm intervals with the 0-40

cm soil layer three days after the final infiltration. Soil $BD$ was calculated using the soil
core method, and $TP$ was calculated by Eq. (3) based on $BD$. Soil water characteristics
curve was measured with the high velocity centrifugal method (CR21 Hitachi, Japan),
and calibrated by RETC software (PC Progress Inc., Prague, Czech Republic).
Currently, several defining sizes of macropores are proposed, rather than a precise
definition and pore size range (Cameira et al. 2003; Kim et al. 2010; Hu et al. 2018;
Budhathoki et al. 2022; Aldaz-Lusarreta et al. 2022).  In this study, macropores were
defined as the pores with diameters larger than 1 mm, whereas micropores were defined
as smaller than 1 mm (Luxmoore 1981; Wilson and Luxmoore 1988). Based on the
capillary pressure data, the relationship between pore diameter ($d$, mm) and water
suction ($S$, Pa) was described according to the capillary bundle model (Jury et al., 1991):





$$d = \frac{300}{S}$$
(7)

**2.4 Statistical analysis**

206 Statistical analysis among all treatments with different $K^+/Na^+$ was performed in

SPSS 22.0 software, using one-way analysis of variance (ANOVA) based on the least
significant difference (LSD) test at 95% significance level ($P < 0.05$). All figures were
created through Origin 2022b.
**3 Results**
**3.1 Soil saturated hydraulic conductivity ($K_{sat}$)**

212 The K0Na1, K1Na1 and K1Na0 significantly ($P < 0.05$) reduced $K_{sat}$ by 43.62%,

29.04% and 18.06% compared with CK, respectively (Fig. 2). Additionally, $K_{sat}$ was
negatively correlated with CROSS of saline water, increasing the CROSS of the applied
saline water generally reduced $K_{sat}$.





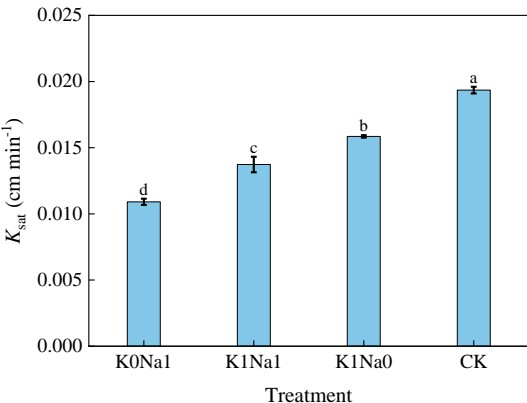


Fig. 2. Saturated hydraulic conductivity ($K_\mathrm{sat}$) under different treatments. K0Na1,
K1Na1 and K1Na0 indicate the saline water at EC of 4 dS m$^{-1}$ with K$^+$/Na$^+$ of 0:1, 1:1
and 1:0, respectively; CK, deionized water; Different letters after means of $K_\mathrm{sat}$ indicate
statistical differences (P < 0.05) among treatments based on LSD. Bars indicate
standard deviations of means.
**3.2 Soil water content**
Water content increased immediately after each infiltration for all treatments, and
then gradually declined to a constant level (Fig. 3). And water content at deeper soil
depths was greater than at shallow soil depths at the same time during the whole
infiltration period. The water content ranged from 0.39-0.41 and 0.40-0.42 cm$^3$ cm$^{-3}$ at
15 and 30 cm soil depths, respectively. K0Na1 gained the highest water content at both
15 and 30 cm soil depths. K1Na1 and K1Na0 were greater than CK at 15 cm soil depth
and lower than CK at 30 cm soil depth, and the water content of K1Na1 was higher
than K1Na0 at both 15 and 30 cm soil layers.



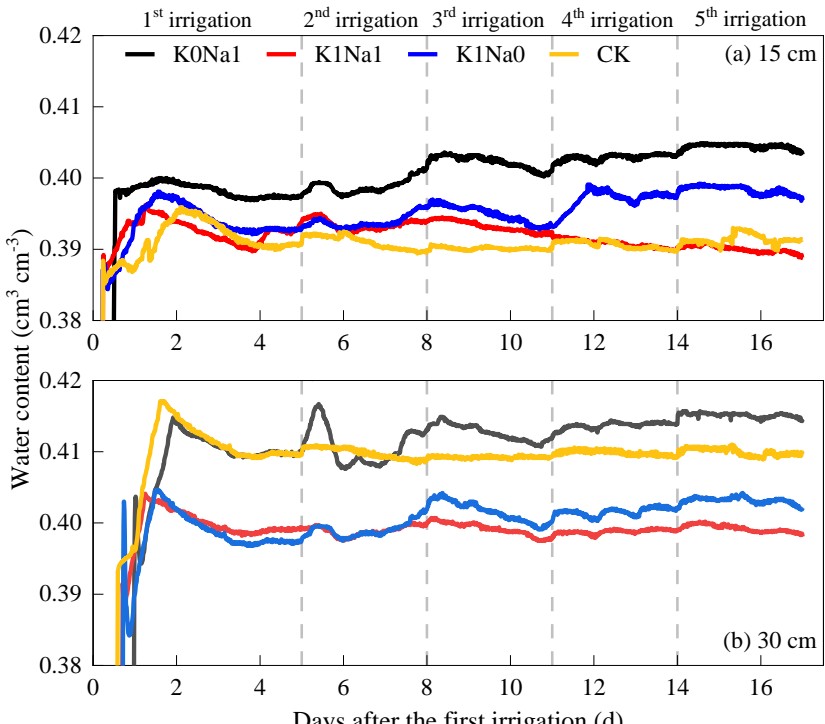


Fig. 3. Variation of water content over time under different treatments at 15 (a) and 30

cm (b) soil depths during the five times of infiltration.

**3.3 Soil salination rate (*Rs*) and leaching efficiency (*Le*)**

The *Rs* and *Le* under CK were not shown in Fig. 4, because deionized water was

used as the control and there was almost no salt contained in the study soil, CK was

considered negligible for salt accumulation and leaching. *Rs* peaked at the first

infiltration, and approximately 70%-80% of the salt in the saline water was retained in

the soil column, after which the subsequent leaching had lower *Rs* values (Fig. 4). The

lower the ratio of $K^+/Na^+$, the larger soil *Rs*. Among the three saline water treatments,





K1Na0 had the lowest *Rs* and highest *Le* at five infiltrations.

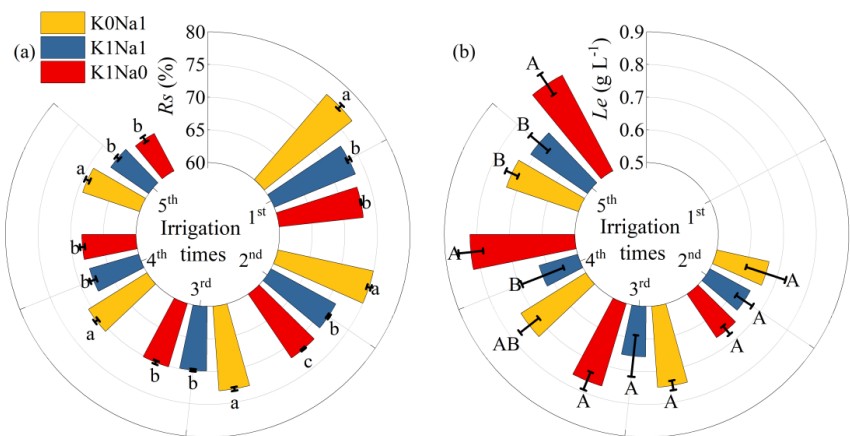


Fig. 4. Salination rate (*Rs*) (a) and leaching efficiency (*Le*) (b) at five infiltrations under
all saline water treatments. K0Na1, K1Na1 and K1Na0 indicate the saline water at EC
of 4 dS m$^{-1}$ with K$^+$/Na$^+$ of 0:1, 1:1 and 1:0, respectively; Different lowercase letters
followed means of *Rs* indicate statistical differences (P < 0.05) among treatments based
on LSD, and different capital letters followed means of *Le* indicate statistical
differences (P < 0.05) among treatments based on LSD. Bars indicate standard
deviations of means.
**3.4 Soil bulk electrical conductivity**

Bulk electrical conductivity of K0Na1, K1Na1 and K1Na0 ranged from 1.0 to 2.0

dS m$^{-1}$ at 15 cm, 1.5 to 2.5 dS m$^{-1}$ at 30 cm soil depth (Fig. 5). After the first infiltration,
bulk electrical conductivity in 15 cm soil layer reached its apes, and then exhibited a
general downward trend in the following infiltrations. However, more salts were





leached to deeper layers, where salt began to accumulate instead of desalination, and
bulk electrical conductivity at 30 cm soil depth gradually increased following the
infiltration events. At both 15 and 30 cm soil layers, the bulk electrical conductivity of
K0Na1 was considerably greater than K1Na1, and K1Na1 was quite higher than K1Na0.

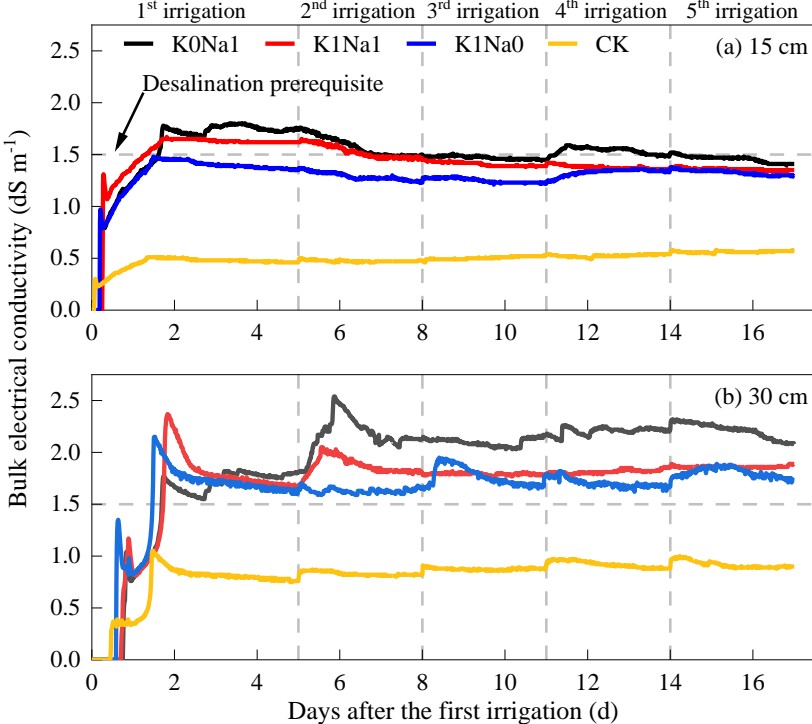


Fig. 5. Variation of bulk electrical conductivity over time under treatments with
different $K^+/Na^+$ at constant EC at 15 (a) and 30 cm (b) soil depths in the period of five
times of infiltration. K0Na1, K1Na1 and K1Na0 indicate the saline water at EC of 4 dS
$m^{-1}$ with $K^+/Na^+$ of 0:1, 1:1 and 1:0, respectively; CK, deionized water.
At 15 cm soil depth, K0Na1 reached the soil desalination prerequisite after the
fifth infiltration, while K1Na1 reached the desalination prerequisite after the second
infiltration, and K1Na0 did not exceed desalination prerequisite during the whole





infiltration period. Among all saline water treatments, K1Na0 saved the desalination
time and required leaching volume to reach the standard of desalination. K0Na1,
K1Na1 and K1Na0 did not meet the desalination prerequisite at 30 cm soil depth, and
the increased volume of infiltration water also increased the bulk electrical conductivity.
**3.5 Soil bulk density (*BD*) and total porosity (*TP*)**

Soil *BD* varied between 1.30 and 1.40 g cm$^{-3}$ for all treatments, and *BD* was below

1.35 g cm$^{-3}$ at 0-10 and 35-40 cm soil layers, however, over 1.35 g cm$^{-3}$ at 10-35 cm
soil depth (Fig. 6). K0Na1 significantly ($P < 0.05$) enhanced soil *BD* throughout the soil
column profile compared with CK. *TP* first diminished with soil depth to reach a
minimum at about 30-35 cm, and then slightly increased at 35-40 cm. The *TP* of K1Na1
and K1Na0 slightly improved after five times of infiltration, and only K0Na1 showed
a decline compared with CK. Overall, over the whole infiltration period, K1Na0 was
most conducive to the formation of soil pore structure and increasing the total pore
volume. The saline water with lower CROSS was beneficial for reducing soil *BD* and
increasing *TP*.



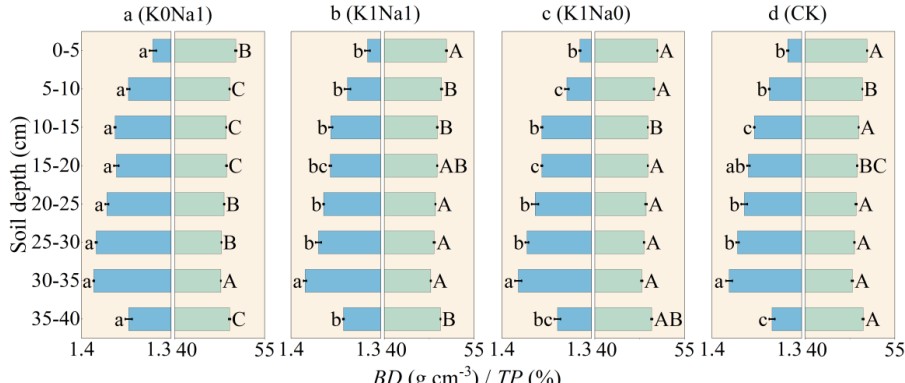


Fig. 6. Soil bulk density (*BD*) and total porosity (*TP*) throughout the soil column profile

under different treatments. K0Na1, K1Na1 and K1Na0 indicate the saline water at EC

of 4 dS m$^{-1}$ with K$^+$/Na$^+$ of 0:1, 1:1 and 1:0, respectively; CK, deionized water; The

blue horizon columns represent *BD*, while the green horizon columns represent *TP*;

Different lowercase letters followed means of *BD* indicate statistical differences (P <

0.05) among treatments based on LSD, and different capital letters followed means of

*TP* indicate statistical differences (P < 0.05) among treatments based on LSD. Bars

indicate standard deviations of means.

**3.6 Proportion of micropores and proportion of macropores**

Micropores were the dominant pores for all treatments, the proportion of

micropores accounting for more than 40% of the total soil volume, however, the

proportion of macropores made up no more than 8% (Fig. 7). 0-5 cm soil layer had the

lowest proportion of macropores and retained the largest proportion of micropores

compared with other depths. K0Na1 had the highest proportion of micropores and the





lowest proportion of macropores. K1Na0 had a greater proportion of macropores in the
soil column compared with CK.

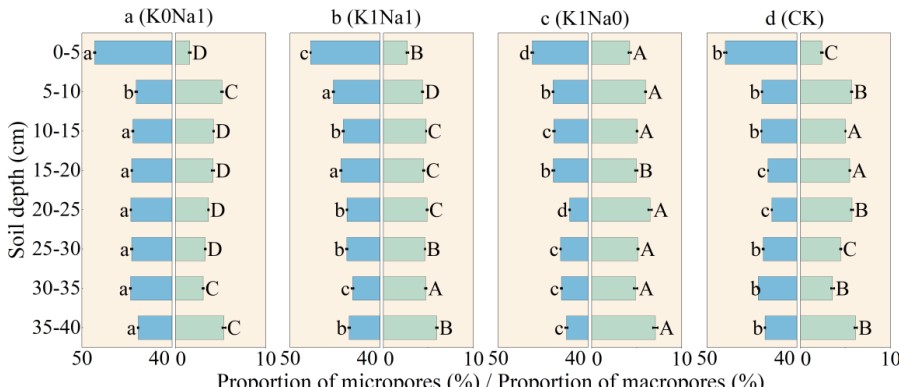


Fig. 7. Proportion of micropores and proportion of macropores in total soil volume
throughout the soil column profile under different treatments. K0Na1, K1Na1 and
K1Na0 indicate the saline water at EC of 4 dS m$^{-1}$ with K$^{+}$/Na$^{+}$ of 0:1, 1:1 and 1:0,
respectively; CK, deionized water; The blue horizon columns represent proportion of
micropores, while the green horizon columns represent proportion of macropores;
Different lowercase letters followed means of proportion of micropores indicate
statistical differences (P < 0.05) among treatments based on LSD, and different capital
letters followed means of proportion of macropores indicate statistical differences (P <
0.05) among treatments based on LSD. Bars indicate standard deviations of means.
**4 Discussion**
**4.1 Effects of saline water on soil water movement and redistribution**

As a crucial soil hydraulic characteristic, $K_{sat}$ represents the transportation ability





of water and solutes (Braud et al. 2001; Maillard et al. 2011; Albalasmeh et al. 2022).
The cation composition and EC of soil solution affect $K_{sat}$ by controlling electrostatic
repulsive pressure through surface potential and midpoint potential between adjacent
particles, and consequently influence water movement (Fares et al. 2000; Liu et al.,
2022b). Specifically, the $K^+/Na^+$ ratio in saline water was related to the swelling and
dispersion of soil particles (Yu et al. 2016; Zhu et al. 2019). Dispersed clay particles
clogged soil macropores to subsequently restrict water transport (Awedat et al., 2021).
The $Na^+$ has a relatively higher ionicity index than $K^+$, as a result, the low $K^+/Na^+$ ratio
decreased the degree of covalency in clay-cation bonds, which was detrimental to clay
particles aggregation (Marchuk and Rengasamy 2011). Therefore, in our study, the high
$K^+/Na^+$ ratio promoted the flocculation and stabilization of soil clay particles, resulting
in an increased infiltration rate.

After a certain period of water supply, soil moisture redistributed at different

depths of soil column. Soil water moved further down during the phase of water
redistribution soon after each irrigation, reducing the water content in the upper soil
layers. As the upper soil layers drained, the lower soil layers still had water inflow
(Kargas et al., 2021), increasing the water content in the lower soil layers. The results
also implicated that the retention of soil water by $Na^+$ was stronger than that by $K^+$, the
cause may be that $Na^+$ can increase the thickness of the diffuse-double layers around
soil colloids theoretically due to its larger hydrated radius and lower charge than $K^+$,
and the adjacent double layers overlapped to provide more space between layers, where,




subsequently, more water can be retained (He et al., 2015). Additionally, our study
showed that an appropriate concentration of $K^+/Na^+$ was even more beneficial than
deionized water for water downward transport, which could be because the deionized
water (CK) (below 0.2 dS m$^{-1}$) tended to leach soluble minerals and salts, especially
$Ca^{2+}$, from the surface soil layers. This would lead to the reduction of its original solid
soil structural stability. In the absence of salt and $Ca^{2+}$, the dispersed tiny particles filled
the smaller pore spaces in soil, reducing even more channels for water flow and
exacerbating water retention in deeper soil layers (Ayers and Westcot 1985). However,
a lower concentration of soluble salts could increase colloid flocculation,  and thereby,
improve soil aeration and water conductivity (Tang and She 2016).

**4.2 Effects of saline water on soil salination and desalination process**

Numerous factors influence the leaching efficiency of soil salts; for example,
increasing EC and reducing SAR definitely improve clay flocculation, which can
enhance salt leaching (Ebrahim Yahya et al., 2022). $Na^+$ is more likely to trigger soil
clay dispersion and swelling than $K^+$, thus $Na^+$ generally inhibits water infiltration,
which is detrimental to salt leaching (Smiles and Smith 2004). Adding $K^+$ could
promote displacement of the adsorbed $Na^+$, and then decrease $Na^+$ concentration and
salt accumulation in soil solute through leaching.
A greater reduction in $Na^+$ concentration was associated with a higher rate of
cation exchange rate, and the slow rate of solute leaching from aggregates reduced the



total leaching efficiency (Shaygan et al., 2017). During the leaching process, water flow
preferentially passed through the macropores rather than aggregates. The slow water
transportation through aggregates induced the slow removal of solutes from the
aggregates, leading to a reduced leaching efficiency. In our study, the alternate leaching
was implemented to improve solute leaching. The soil solutes diffused into the
aggregates surface during the rest period, improving salt leaching due to the water flow
in macropores (Al-Sibai et al., 1997). Increasing $K^+/Na^+$ ratio could increase the
magnitude of cation exchange due to the substitution of $Na^+$ on exchange sites by $K^+$
with lower dispersive potential (Shaygan et al., 2017), the intensive release of cations
from the soil further improved salt's leaching efficiency. In addition, the integrity of
soil aggregates created by combining clay particles and the other soil components
enhanced by $K^+$ can benefit solute transportation (Marchuk and Rengasamy 2011).
**4.3 Effects of saline water on soil pore structure characteristics**

The upper soil was longer exposed to water due to the long-term continuous

irrigation, causing the particles to swell and the surface layer to loosen (Vaezi et al.
2017; Håkansson and Lipiec 2000), and also the decreased *BD* in the surface layer of
soil column. The subsoil *BD* increased with depth under the impact of water pressure
and self-weight due to the declining pore diameter and pore branching closure
(Schjønning et al., 2013). And for soil at the bottom, the loss of soil particles from small
holes was responsible for the abrupt reduction in *BD*. The value of CROSS in saline





water could reflect changes in soil *BD* and *TP*, in agreement with the result of Marchuk
and Marchuk (2018). The high CROSS implied an increase in the proportion of
monovalent exchangeable cations, thickening the double layer at the interface between
the clay surface and soil solution. Hence, soil swelling occurred at the expense of water-
conducting pores. Additionally, aggregates slaking and subsequent clay dispersion and
deposition of clay particles within the pore space contributed to the reduction in *TP*
(Marchuk and Marchuk 2018).
Fewer soil macropores plays a crucial role in water and solute transport,
accounting for 85% of the total infiltration volume (Wilson and Luxmoore 1988; Weiler
and Naef 2003; Kotlar et al. 2020). The lower the $K^+/Na^+$ ratio, the more it enhanced
soil clay dispersion, resulting in the loosening of clay particles from the aggregates.
This, in turn, dispersed clay particles moved with water caused the macropores to
become blocked, converting them into micropores (Cameira et al., 2003), thus leading
to a decrease in the volume of soil macropores.
**5 Conclusion**
We explored the effect of $K^+/Na^+$ in saline water on soil hydraulic characteristics
and structural stability via a soil column experiment. The higher $K^+/Na^+$ ratio caused
fewer pore blockages due to soil clay particle dispersion than lower $K^+/Na^+$, which
increased soil saturated hydraulic conductivity. The presence of $K^+$ accelerated the
sustained $Na^+$ replacement and leaching, alleviating salt accumulation and enhancing



leaching efficiency. Increasing $K^+/Na^+$ positively affected the establishment of soil
structure due to the transformation of micropores into macropores, and the ever-
increasing unobstructed water-conducting channels sped up water and solute transport.
The rational use of saline water with adequate $K^+$ could help mitigate the structural
deterioration caused by $Na^+$. Appropriate adjustment of saline water $K^+/Na^+$ during
infiltration could ameliorate soil structural properties. In addition to $Ca^{2+}$ and $Mg^{2+}$
(primary concerns in earlier studies), the relative concentration of $K^+$ to $Na^+$ is an
essential indicator for assessing the suitability of saline water quality for irrigation and
should be considered when using saline water.
**Author contributions**

Sihui Yan and Tibin Zhang conceived and designed the experiments. Sihui Yan,

Binbin Zhang and Tonggang Zhang led the data processing and statistical analysis,
Sihui Yan, Yu Cheng, Chun Wang and Min Luo performed the experiments. Sihui Yan
wrote the initial draft. Hao Feng and Kadambot H.M. Siddique contributed to review
and editing of the paper.
**Acknowledgments**

This work was supported by the National Key R&D Program of China (Grant No.

2021YFD1900700) and National Natural Science Foundation of China (Grant No.

51879224, 51509238).



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
