# Peer review of "The higher relative concentration of K+ to Na+ in saline"

_EGUsphere, 2022_

## Author Comment (AC1)

Dear reviewer:

Thank you for providing many constructive suggestions concerning our manuscript (EGUSPHERE-2022-1390) entitled "The higher relative concentration of $K^+$ to $Na^+$ in saline water improves soil hydraulic conductivity, salt leaching efficiency and structural stability". Here are our specific responses to your comments.

If any questions, please don't hesitate to let us know, and we would like to have more discussions with you!

Thank you for your consideration.

Sihui Yan, Tibin Zhang, and on behalf of all authors

**Review 1**

In 1999, I performed a column study of almost equal design on 3 chernozem soils pH 6,8-8,6), together with a master student. Our question was to simulate the vertical transport of a single addition of NPK fertilizer solution by rainwater, containing sulfate and phosphate as anions, beneath anionic trace elements. Addition of $K^+$ led to the release of all other soluble cations, like Na, Li, Mg, Ca, Sr, Ba and $H^+$ (acidification!). In this work, the goal is the substitution of sea water intrusion by KCl fertilizer, which is not so clear from the abstract.

It is well known that clay minerals exert stronger affinity to $K^+$ and $NH_4^+$ than to $Na^+$ and others.

**Question 1:** From the experimental part, it is not quite clear that obviously the NaCl and KCl solutions (of equal conductivity) had not been added to columns run in parallel, but subsequently to the same columns (I took this from figures 3 and 5).
**Response:** Our soil column experiments were set up with different sodium to potassium ratios at the same EC level, and saline water was infused into the soil column in five times. Alternate leaching was formed to provide enough time for cation leaching, water

redistribution and pore structure changes (Lines 150-153). The soil column was saturated after the first irrigation, and subsequent irrigations were also with saline water of the same treatment. The addition of saline water increases the salt content and also promotes salt leaching. We, therefore, wished to simulate the effect of different sodium-to-potassium ratios of saline water on soil water-salt transport and macropore distribution under such irrigation conditions.

**Question 2:** 6 liters of approximate pore volume is realistic. But how much salt solution was added, and was there a wash-down with water in between?

**Response:** We calculated the pore volume in the soil to be 6 L. The first irrigation was expected to saturate the soil, i.e. the irrigation water filled all the pores in the soil. Therefore, our irrigation volume was the calculated pore volume of 6 L. However, because of the limitation of the soil water holding capacity, some of the water was leached outside of the soil column, and the volume of leachate was about 0.5 L. After 2 days after the end of the leaching (the end of the redistribution of soil water), to keep the soil saturated, we irrigated the soil with the same saline water as the first irrigation solution, with a volume of 0.5 L, and the subsequent irrigation was repeated four times (Lines 166-169). Thus, there was no wash-down in between.

**Question 3:** What had been measured to obtain the amount of released salt?

**Response:** To determine the amount of salt released, we measured the volume and electrical conductivity of the leachate. The leachate was collected at 3 h intervals when the leachate started to drain. The leachate was stored in 100 ml wide-mouth polypropylene reagent bottles, volume and electrical conductivity of the leachate were measured. Maybe this fact was not clearly expressed in the article, and we will do a detailed revision of the article later. Thank you for your reminder.

**Question 4:** Only Na and K, but also Ca, Mg, Sr, Ba, sulfate, carbonate? These data are completely missing!

**Response:** The experiment you did in 1999 was very comprehensive and took into

account many aspects, and your study was very informative for our experiment. We did this experiment because, first of all, $K^+$ does affect $Na^+$ leaching, however, the presence of $Na^+$ could still cause soil clay particle dispersion, which thus inhibits water infiltration and prevent salt leaching. Secondly, the change of soil pore structure during water redistribution could affect the rate of soil water and salt transport, which are factors affecting $Na^+$ leaching. We, therefore, set up this experiment to analyze the combined effects of $K^+$ and $Na^+$ interactions on soil water and salt movement. In order to exclude the effect of other factors, the ion concentration contained in the soil we chose was very low (Table 1), so the effects of $K^+$ and $Na^+$ on other ions were not considered. Thank you very much for your suggestion, we should do more related research in this respect in the future study, and we hope we can keep the academic communication with you in the future.

**Question 5:** (Line 212) The abbreviation Ksat is misleading, because this is not saturated potassium, but saturated hydraulic conductivity!

**Response:** The fact that Ksat is defined as the abbreviation of Saturated Hydraulic Conductivity of Soil is widely accepted in the field of soil physics. Based on this fact, we still prefer to use this expression.

**Question 6:** (Fig. 2) The given saturated hydraulic conductivity obviously starts from differently pretreated soils, if the test solutions had been added to the same column subsequently. My main discipline is analytical chemistry, and not soil physics - no comments upon hydraulic parameters

**Response:** The saturated hydraulic conductivity, as shown in Figure 2, was determined on the soil after different pretreatments. Initially, all physicochemical properties and external conditions of the soil were the same, and we assumed that the original saturated hydraulic conductivity of all soils was the same. After saline water irrigation, the action of cations in the soil and changes in the pore structure affected the saturated hydraulic conductivity, so we determined the saturated hydraulic conductivity with different pretreatments. Thank you again for your comments.

---

## Author Comment (AC2)

Dear reviewer:

Thank you for providing many constructive suggestions. Replacing NaCl in the pore water by $K_2SO_4$ is really a good suggestion, because $SO_4^{2-}$ is less damaging to the soil structure compared to $Cl^{-1}$, so this might be a better way to assess the role of different cations.

Thanks again.

---

## Author Comment (AC3)

Dear reviewer:

Thank you for providing many constructive suggestions concerning our manuscript (EGUSPHERE-2022-1390). Here are our specific responses to your comments.

If any questions, please let us know, and we would like to have more discussions with you.

Sihui Yan, Tibin Zhang, and on behalf of all authors

**Question 1:** Introduction: Line 53: 'this disaster', the phrase to describe saline water irrigation leading to soil salinization may not be suitable. I suggest the authors to correct it.

**Response:** Thank you for your reminder, we will correct this word to 'This phenomenon is related directly to soil pore size distribution, and in turn to the dispersion and swelling of the clay fraction.'

**Question 2:** Line 53-55: how the soil salinization is related to the pore size distribution. It needs more explanation or description here.

**Response:** Cations in saline water could cause soil clay particles to disperse or flocculate to affect changes in soil pore structure, and pore distribution could influence the fluidity of soil water flow channels, which can affect soil water and solute transport and alter soil salinization. We will add more specific descriptions and explanations in the paper.

**Question 3:** Lines 55-57: I did not see any background about the urgency to study the effects of saline water quality on soil hydraulic properties.

**Response:** We will add background on soil hydraulic properties before this sentence and add this description: Once the soil is salinized and/or alkalized, soil hydraulic properties, like infiltration rate, saturated hydraulic conductivity and permeability, will change inevitably. Therefore, in order to optimize saline water utilization, the effects of

saline water quality on the soil hydraulic properties and pore structure characteristics should be paid more attention.

**Question 4:** Line 60: 'Clay exchange surface', here is missing the key element that, 'clay' does not exchange but the ions on the clay surface will.

**Response:** This sentence is revised as: Excess sodium ($Na^+$) from irrigation saline water is adsorbed onto the clay surface in salt-affected soils where sodium compounds predominate contributing to the disintegration of soil structure.

**Question 5:** Line 61-62: Please add more information about how the soil structure would be disintegrated due to the predominate sodium.

**Response:** As percolation progresses, the thickness of the diffusion double electron layers raises due to the relatively larger hydrated radius of $Na^+$, and the repulsive force between adjacent diffusion double electron layers appears to increase, resulting in the dispersion and swelling of soil particles (Alva et al. 1991; Reading et al. 2015), causing soil structure deterioration due to poor soil cementation (Lines 63-66).

**Question 6:** Line 99-100: it was the first time to see the prediction index in the objectives of the paper (CROSS and SAR), a newly proposed index (CROSS)… The authors should shortly introduce them in the introduction.

**Response:** In the introduction, we can add the following description:

Traditional SAR ignored the role of $K^+$, a newly proposed equation, cation ratio of soil structural stability (CROSS) could integrate the effects of $Na^+$ and $K^+$ in soil, which is an important indicator for assessing the quality of brackish water.

**Question 7:** Materials & Methods

Line 109-110: It would be better to provide the content of total salts of the soil to show the salt concentration of soil is low, EC value could not 100% percent to replace the salt

concentration.

**Response:** We will add information about the amount of salt added.

**Question 8:** Fig 1: From the Fig 1a, I did not see the bottom of apparatus, for instance the part which connecting leachate catcher.

**Response:** The silver device at the bottom edge of the soil column is the funnel that collects the leachate, but the lower funnel exit was not captured due to the camera view, so we added Figure 1b schematic to show it more visually.

**Question 9:** Section '2.3 Measurements': please be more specific about the section title.

**Response:** We could use 'Soil properties measurements' as the title.

**Question 10:** Results: Section '3.5 Soil bulk density (BD and total porosity (TP))': I questioned the data of soil BD in this section, if the authors used cutting ring method to obtain the soil BD data. The height of the cutting ring is about 5 cm, which was exactly the interval of the soil depth (5 cm), then how could the authors to manage the cutting ring sampling to make sure involve enough soil in the rings?

**Response:** Yes, we used the cutting ring method to obtain the soil BD. The fact is that the diameter of the soil column is 20 cm and the area is about 314 $cm^2$. The cross-sectional area of the column is very large; besides, we avoid sampling soil at the same location for the different soil layers during the experiment.

**Question 11:** Instead of SAR, using the new prediction index CROSS was one of the main objectives in the manuscript, I suppose to investigate the comparison between these two indexes. However, such a part of information was lack in the results section and discussion section.

**Response:** Since K0Na1 has no added $K^+$, the potassium adsorption ratio is 0, and K1Na0 has no added $Na^+$, the sodium adsorption ratio is 0. Neither the SAR nor the PAR can be analyzed to characterize the cation composition, so we chose CROSS. the superiority of CROSS over SAR has been corroborated by many previous studies, so

the differences between SAR and CROSS are not analyzed separately in this paper.

**Question 12:** Discussion: Line 338, line 363, line 386: Based on the experiment design, it only had two ratios of the $K^+/Na^+$, which were 1 and 0. In this case, it could not show too much evidence from this manuscript about the ratio of these two ions affecting either soil pores or bulk density. Try to use other ways or change a perspective to discuss the effect of $K^+$ and $Na^+$ on soil structure.

**Response:** We could revise these sentences to:

Additionally, our study showed that K1Na1 was even more beneficial than deionized water for water downward transport.

Saline water with more $K^+$ could increase the magnitude of cation exchange due to the substitution of $Na^+$ on exchange sites by $K^+$ with lower dispersive potential.

For saline water with the same electrolyte concentration, a decrease in $K^+$ concentration may enhance soil clay dispersion, resulting in the loosening of clay particles from the aggregates.

**Question 13:** Conclusions: As I proposed the comment for the discussion, I would guess the effect of the ratio or the relative concentrations of $K^+$ and $Na^+$ on saline water irrigation to soil, however, at the current version of the manuscript, it would be better to consider other way to conclude this.

**Response:** Thank you very much for your suggestion, the conclusion section could be rephrased as follows:

We explored the effect of the ratio of $K^+$ to $Na^+$ in saline water on soil hydraulic characteristics and structural stability via a soil column experiment. Irrigation with saline water of $K^+/Na^+$ of 1:0 caused fewer pore blockages due to soil clay particle

dispersion than 0:1, which increased soil saturated hydraulic conductivity. The presence of $K^+$ accelerated the sustained $Na^+$ replacement and leaching, alleviating salt accumulation and enhancing leaching efficiency. $K^+$ positively affected the establishment of soil structure due to the transformation of micropores into macropores, and the ever-increasing unobstructed water-conducting channels sped up water and solute transport. The rational use of saline water with adequate $K^+$ could help mitigate the structural deterioration caused by $Na^+$. Appropriate adjustment of the relative concentration of $K^+$ to $Na^+$ in saline water during infiltration could ameliorate soil structural properties. In addition to $Ca^{2+}$ and $Mg^{2+}$ (primary concerns in earlier studies), the relative concentration of $K^+$ to $Na^+$ is an essential indicator for assessing the suitability of saline water quality for irrigation and should be considered when using saline water.

---

## Author Comment (AC4)

Dear reviewer:

Thank you for providing many constructive suggestions concerning our manuscript (EGUSPHERE-2022-1390). Here are our specific responses to your comments.

If any questions, please let us know, and we would like to have more discussions with you!

Sihui Yan, Tibin Zhang, and on behalf of all authors

**Question 1:** The paper is of interest for the area of saline water management. The methods are generally satisfactory and the paper is generally well organized. -Fig.1: The legend must be completed (figures must be self-explanatory).

**Response:** Thank you for your suggestion, we will add relevant legends to the diagram for a more visual presentation.

[Figure]

Fig. 1. Illustration of the experiment apparatus (a) and schematic diagram (b).

**Question 2:** What about the statistical analysis in Figures 3 and 5?

**Response:** The amount of data for soil moisture and bulk electrical conductivity is very large, and if we do a significance analysis, we can only use the average value, or the value at a point in time after the end of the irrigation cycle, and this does not accurately

represent the process of soil moisture or bulk electrical conductivity, so we did not do a significance analysis for soil moisture or bulk electrical conductivity in Fig.3 and 5.

**Question 3:** Fig. 3: What is the explanation for soil moisture at 15 cm being lower than at 30 cm after an irrigation event? (it is supposed to be higher at 15 cm immediately after an irrigation event)

**Response:** The data analyzed in Fig. 3 and Fig. 5 were started at the time after a certain period of water supply, soil moisture was redistributed at different depths of soil column. Soil water moved further down during the phase of water redistribution soon after each irrigation, reducing the water content in the upper soil layers. As the upper soil layers drained, the lower soil layers still had water inflow (Kargas et al., 2021), increasing the water content in the lower soil layers. (Lines 328-332). And after each irrigation, soil moisture rose rapidly in both 15 cm and 30 cm soil layers.

**Question 4:** Did not started the irrigation events in parallel for the different treatments?

**Response:** The starting point of irrigation was the same for all treatments.

**Question 5:** L226: "Water content increased immediately after each infiltration for all treatments, and then gradually declined to a constant level…" -It does not seem that soil water content has become constant (Fig. 3).

**Response:** We can rephrase this sentence to more accurately describe: 'Water content increased immediately after each infiltration for all treatments, then gradually decreases and the degree of variation tends to stabilize'.

**Question 6:** L260: "At both 15 and 30 cm soil layers, the bulk electrical conductivity of K0Na1 was considerably greater than K1Na1, and K1Na1 was quite higher than K1Na0." -I do not took this information from Fig. 5, particularly at 15 cm.

**Response:** We can revise the sentence to: Overall, K0Na1 had the highest bulk electrical conductivity among all treatments at both 15 and 30 cm, and K1Na1 was quite higher than K1Na0.

**Question 7:** L267: "At 15 cm soil depth, K0Na1 reached the soil desalination prerequisite…" -What could be the reason for the increase in bulk electrical conductivity at 15 cm in K0Na1 after 4th irrigation?

**Response:** The decrease in the soil macro-porosity, soil water retention, and weaker hydraulic conductivity all contribute to the increase in bulk electrical conductivity of K0Na1. A greater reduction in $Na^+$ concentration was associated with a higher rate of cation exchange rate, and the slow rate of solute leaching from aggregates reduced the total leaching efficiency (Shaygan et al., 2017). During the leaching process, water flow preferentially passed through the macropores rather than aggregates. The slow water transportation through aggregates induced the slow removal of solutes from the aggregates, leading to a reduced leaching efficiency. In our study, the alternate leaching was implemented to improve solute leaching. The soil solutes diffused into the aggregates surface during the rest period, improving salt leaching due to the water flow in macropores (Al-Sibai et al., 1997). Increasing the relative ratio of $K^+$ to $Na^+$ could increase the magnitude of cation exchange due to the substitution of $Na^+$ on exchange sites by $K^+$ with lower dispersive potential (Shaygan et al., 2017), the intensive release of cations from the soil further improved salt's leaching efficiency. In addition, the integrity of soil aggregates created by combining clay particles and the other soil components enhanced by $K^+$ can benefit solute transportation (Marchuk and Rengasamy 2011) (Lines 355-368).

**Question 8:** L326: "Therefore, in our study, the high $K^+/Na^+$ ratio promoted the flocculation and stabilization of soil clay particles, resulting in an increased infiltration rate." -What about infiltration rate data? There is no information in the manuscript.

**Response:** This sentence illustrates the data in Fig. 2 (Saturated hydraulic conductivity under different treatments), and to avoid misleading, we change this sentence to: Therefore, in our study, the high relative concentration of $K^+$ to $Na^+$ promoted the flocculation and stabilization of soil clay particles, resulting in an increased water hydraulic conductivity.

**Question 9:** I suggest indicating the number of figure/table throughout the discussion (e.g. L332 "The results also implicated…").

**Response:** We will indicate the figure and table number in the discussion so that it can be more clearly represented.

---

## Author Response (AR1)

Dear Editor and Reviewers:

Thank you for providing many constructive suggestions concerning our manuscript (EGUSPHERE-2022-1390) entitled "The higher relative concentration of $K^+$ to $Na^+$ in saline water improves soil hydraulic conductivity, salt leaching efficiency and structural stability". Here are our specific responses to your comments. Corrections were made based on the recommendations, and detailed response to each comment is provided. We are looking forward to receiving your feedback. We would be pleased if the paper is selected for publication in the Soil.

If any questions, please don't hesitate to let us know.

Thank you for your consideration.

Sihui Yan, Tibin Zhang, and on behalf of all authors

**Reviewer 1**

**Question 1:** In 1999, I performed a column study of almost equal design on 3 chernozem soils pH 6,8-8,6), together with a master student. Our question was to simulate the vertical transport of a single addition of NPK fertilizer solution by rainwater, containing sulfate and phosphate as anions, beneath anionic trace elements. Addition of $K^+$ led to the release of all other soluble cations, like Na, Li, Mg, Ca, Sr, Ba and $H^+$ (acidification!). In this work, the goal is the substitution of sea water intrusion by KCl fertilizer, which is not so clear from the abstract.

It is well known that clay minerals exert stronger affinity to $K^+$ and $NH_4^+$ than to $Na^+$ and others.

From the experimental part, it is not quite clear that obviously the NaCl and KCl solutions (of equal conductivity) had not been added to columns run in parallel, but subsequently to the same columns (I took this from figures 3 and 5).

**Response:** Our soil column experiments were set up with different $K^+$ to $Na^+$ ratios at the same EC level, and saline water was infused into the soil column in five times.

Alternate leaching was formed to provide enough time for cation leaching, water redistribution and pore structure changes (Lines 150-152). The soil column was saturated after the first irrigation, and subsequent irrigations were also with saline water of the same treatment. The addition of saline water increases the salt content and also promotes salt leaching. We, therefore, wished to simulate the effect of different $K^+$ to $Na^+$ ratios of saline water on soil water-salt transport and macropore distribution under such irrigation conditions.

**Question 2:** 6 liters of approximate pore volume is realistic. But how much salt solution was added, and was there a wash-down with water in between?

**Response:** We calculated the pore volume in the soil to be 6 L. The first irrigation was expected to saturate the soil, i.e. the irrigation water filled all the pores in the soil. Therefore, our irrigation volume was the calculated pore volume of 6 L. However, because of the limitation of the soil water holding capacity, some of the water was leached outside of the soil column, and the volume of leachate was about 0.5 L. After 2 days after the end of the leaching (the end of the redistribution of soil water), to keep the soil saturated, we irrigated the soil with the same saline water as the first irrigation solution, with a volume of 0.5 L, and the subsequent irrigation was repeated four times (Lines 165-168). Thus, there was no wash-down in between.

**Question 3:** What had been measured to obtain the amount of released salt?

**Response:** To determine the amount of salt released, we measured the volume and EC of the leachate. The leachate was collected at 3 h intervals when the leachate started to drain. The leachate was stored in 100 ml wide-mouth polypropylene reagent bottles (Lines 181-183).

**Question 4:** Only Na and K, but also Ca, Mg, Sr, Ba, sulfate, carbonate? These data are completely missing!

**Response:** The experiment you did in 1999 was very comprehensive and took into

account many aspects, and your study was very informative for our experiment. We did this experiment because, first of all, $K^+$ does affect $Na^+$ leaching, however, the presence of $Na^+$ could still cause soil clay particle dispersion, which thus inhibits water infiltration and prevent salt leaching. Secondly, the change of soil pore structure during water redistribution could affect the rate of soil water and salt transport, which are factors affecting $Na^+$ leaching. We, therefore, set up this experiment to analyze the combined effects of $K^+$ and $Na^+$ interactions on soil water and salt movement. In order to exclude the effect of other factors, the ion concentration contained in the soil we chose was very low (Table 1), so the effects of $K^+$ and $Na^+$ on other ions were not considered. Thank you very much for your suggestion, we should do more related research in this respect in the future study, and we hope we can keep the academic communication with you in the future.

**Question 5:** (Line 212) The abbreviation Ksat is misleading, because this is not saturated potassium, but saturated hydraulic conductivity!

**Response:** The fact that Ksat is defined as the abbreviation of Saturated Hydraulic Conductivity of Soil is widely accepted in the field of soil physics. Based on this fact, we still prefer to use this expression.

**Question 6:** (Fig. 2) The given saturated hydraulic conductivity obviously starts from differently pretreated soils, if the test solutions had been added to the same column subsequently. My main discipline is analytical chemistry, and not soil physics - no comments upon hydraulic parameters

**Response:** The saturated hydraulic conductivity, as shown in Figure 2, was determined on the soil after different pretreatments. Initially, all physicochemical properties and external conditions of the soil were the same, and we assumed that the original saturated hydraulic conductivity of all soils was the same. After saline water irrigation, the action of cations in the soil and changes in the pore structure affected the saturated hydraulic conductivity, so we determined the saturated hydraulic conductivity with different

pretreatments. Thank you again for your comments.

**Reviewer 2**

**Question 1:** Introduction: Line 53: 'this disaster', the phrase to describe saline water irrigation leading to soil salinization may not be suitable. I suggest the authors to correct it.

**Response:** Thank you for your reminder, we have revised this sentence to 'Cations in the soil solution change the soil structural characteristics through the soil clay particle dispersion and flocculation' (Lines 55-56).

**Question 2:** Line 53-55: how the soil salinization is related to the pore size distribution. It needs more explanation or description here.

**Response:** We have added the explanation (Lines 55-56): cations in the soil solution change the soil structural characteristics through the soil clay particle dispersion and flocculation'.

**Question 3:** Lines 55-57: I did not see any background about the urgency to study the effects of saline water quality on soil hydraulic properties.

**Response:** We have added background on soil hydraulic properties before this sentence and added this description: Once the soil is salinized and/or alkalized, soil hydraulic properties, like infiltration rate, saturated hydraulic conductivity and permeability, will change inevitably (Scudiero et al., 2017). And cations in the soil solution change the soil structural characteristics through the soil clay particle dispersion and flocculation (Bouksila et al. 2013; Hack-ten Broeke et al. 2016; Zhang et al. 2018). Therefore, in order to optimize saline water utilization, the effects of saline water quality on the soil hydraulic properties and pore structure characteristics should be paid more attention (Lines 53-59).

**Question 4:** Line 60: 'Clay exchange surface', here is missing the key element that,

'clay' does not exchange but the ions on the clay surface will.

**Response:** This sentence has been revised as: Excess sodium ($Na^+$) from irrigation saline water is adsorbed onto the clay surface in salt-affected soils where sodium compounds predominate contributing to the disintegration of soil structure (Lines 61-64).

**Question 5:** Line 61-62: Please add more information about how the soil structure would be disintegrated due to the predominate sodium.

**Response:** As percolation progresses, the thickness of the diffusion double electron layers raises due to the relatively larger hydrated radius of $Na^+$, and the repulsive force between adjacent diffusion double electron layers appears to increase, resulting in the dispersion and swelling of soil particles (Alva et al. 1991; Reading et al. 2015), causing soil structure deterioration due to poor soil cementation (Lines 65-68).

**Question 6:** Line 99-100: it was the first time to see the prediction index in the objectives of the paper (CROSS and SAR), a newly proposed index (CROSS)… The authors should shortly introduce them in the introduction.

**Response:** In the introduction, we have added the following description:

Lines 95-98: Traditional SAR ignored the role of $K^+$, a newly proposed equation, cation ratio of soil structural stability (CROSS) could integrate the effects of $Na^+$ and $K^+$ in soil, which is an important indicator for assessing the quality of saline water (Rengasamy and Marchuk 2011).

**Question 7:** Materials & Methods

Line 109-110: It would be better to provide the content of total salts of the soil to show the salt concentration of soil is low, EC value could not 100% percent to replace the salt concentration.

**Response:** We have added the data for soil total soluble salt (0.14 g Kg$^{-1}$) to Table 1 and modified this sentence to:

Lines 114-116: $EC_e$ and pH were measured using conductivity meter (DDS-307, China) and pH meter (PHS-3C, China), respectively. Total soluble salts refer to the total amount of soluble salts in soil-saturated paste extract.

**Question 8:** Fig 1: From the Fig 1a, I did not see the bottom of apparatus, for instance the part which connecting leachate catcher.

**Response:** The silver device at the bottom edge of the soil column is the funnel that collects the leachate, but the lower funnel exit was not captured due to the camera view, so we added Figure 1b schematic to show it more visually.

**Question 9:** Section '2.3 Measurements': please be more specific about the section title.

**Response:** We could use 'Soil properties measurements' as the title.

**Question 10:** Results: Section '3.5 Soil bulk density (BD and total porosity (TP))': I questioned the data of soil BD in this section, if the authors used cutting ring method to obtain the soil BD data. The height of the cutting ring is about 5 cm, which was exactly the interval of the soil depth (5 cm), then how could the authors to manage the cutting ring sampling to make sure involve enough soil in the rings?

**Response:** Yes, we used the cutting ring method to obtain the soil BD. The fact is that the diameter of the soil column is 20 cm and the area is about 314 cm$^2$. The cross-sectional area of the column is very large; besides, we avoid sampling soil at the same location for the different soil layers during the experiment.

**Question 11:** Instead of SAR, using the new prediction index CROSS was one of the main objectives in the manuscript, I suppose to investigate the comparison between these two indexes. However, such a part of information was lack in the results section

and discussion section.

**Response:** Since K0Na1 has no added $K^+$, the potassium adsorption ratio is 0, and K1Na0 has no added $Na^+$, the sodium adsorption ratio is 0. Neither the SAR nor the PAR can be analyzed to characterize the cation composition, so we chose CROSS. the superiority of CROSS over SAR has been corroborated by many previous studies, so the differences between SAR and CROSS are not analyzed separately in this paper.

**Question 12:** Discussion: Line 338, line 363, line 386: Based on the experiment design, it only had two ratios of the $K^+/Na^+$, which were 1 and 0. In this case, it could not show too much evidence from this manuscript about the ratio of these two ions affecting either soil pores or bulk density. Try to use other ways or change a perspective to discuss the effect of $K^+$ and $Na^+$ on soil structure.

**Response:** We have revised these sentences to:

Lines 338-339: Additionally, our study showed that K1Na1 was even more beneficial than deionized water for water downward transport.

Lines 363-365: Saline water with more $K^+$ could increase the magnitude of cation exchange due to the substitution of $Na^+$ on exchange sites by $K^+$ with lower dispersive potential.

Lines 387-389: For saline water with the same EC, a decrease in $K^+$ concentration may enhance soil clay dispersion, resulting in the loosening of clay particles from the aggregates.

**Question 13:** Conclusions: As I proposed the comment for the discussion, I would guess the effect of the ratio or the relative concentrations of $K^+$ and $Na^+$ on saline water irrigation to soil, however, at the current version of the manuscript, it would be better

to consider other way to conclude this.

**Response:** Thank you very much for your suggestion, the conclusion section has been rephrased as follows:

Lines 393-407: We explored the effect of the ratio of $K^+$ to $Na^+$ in saline water on soil hydraulic characteristics and structural stability via a soil column experiment. Irrigation with saline water of $K^+/Na^+$ of 1:0 caused fewer pore blockages due to soil clay particle dispersion than 0:1, which increased soil saturated hydraulic conductivity. The presence of $K^+$ accelerated the sustained $Na^+$ replacement and leaching, alleviating salt accumulation and enhancing leaching efficiency. $K^+$ positively affected the establishment of soil structure due to the transformation of micropores into macropores, and the ever-increasing unobstructed water-conducting channels sped up water and solute transport. The rational use of saline water with adequate $K^+$ could help mitigate the structural deterioration caused by $Na^+$. Appropriate adjustment of the relative concentration of $K^+$ to $Na^+$ in saline water during infiltration could ameliorate soil structural properties. In addition to $Ca^{2+}$ and $Mg^{2+}$ (primary concerns in earlier studies), the relative concentration of $K^+$ to $Na^+$ is an essential indicator for assessing the suitability of saline water quality for irrigation and should be considered when using saline water.

**Reviewer 3**

**Question 1:** The paper is of interest for the area of saline water management. The methods are generally satisfactory and the paper is generally well organized. -Fig.1: The legend must be completed (figures must be self-explanatory).

**Response:** Thank you for your suggestion, we will add relevant legends to the diagram for a more visual presentation.

[Figure]

Fig. 1. Illustration of the experiment apparatus (a) and schematic diagram (b).

**Question 2:** What about the statistical analysis in Figures 3 and 5?

**Response:** The amount of data for soil moisture and bulk electrical conductivity is very large, and if we do a significance analysis, we can only use the average value, or the value at a point in time after the end of the irrigation cycle, and this does not accurately represent the process of soil moisture or bulk electrical conductivity, so we did not do a significance analysis for soil moisture or bulk electrical conductivity in Figs. 3 and 5.

**Question 3:** Fig. 3: What is the explanation for soil moisture at 15 cm being lower than at 30 cm after an irrigation event? (it is supposed to be higher at 15 cm immediately after an irrigation event)

**Response:** The data analyzed in Fig. 3 and Fig. 5 were started at the time after a certain period of water supply, soil moisture was redistributed at different depths of soil column. Soil water moved further down during the phase of water redistribution soon after each irrigation, reducing the water content in the upper soil layers. As the upper soil layers drained, the lower soil layers still had water inflow (Kargas et al., 2021), increasing the water content in the lower soil layers. (Lines 328-332). And after each irrigation, soil moisture rose rapidly in both 15 cm and 30 cm soil layers.

**Question 4:** Did not started the irrigation events in parallel for the different treatments?

**Response:** The starting point of irrigation was the same for all treatments.

**Question 5:** L226: "Water content increased immediately after each infiltration for all treatments, and then gradually declined to a constant level…" -It does not seem that soil water content has become constant (Fig. 3).

**Response:** We have rephrased this sentence to a more accurately describe: 'Water content increased immediately after each infiltration in all treatments, then gradually decreases and the degree of variation tends to stabilize' (Lines 225-226).

**Question 6:** L260: "At both 15 and 30 cm soil layers, the bulk electrical conductivity of K0Na1 was considerably greater than K1Na1, and K1Na1 was quite higher than K1Na0." -I do not took this information from Fig. 5, particularly at 15 cm.

**Response:** We have revised the sentence: Overall, K0Na1 had the highest bulk electrical conductivity among all treatments at both 15 and 30 cm, and K1Na1 was quite higher than K1Na0.

**Question 7:** L267: "At 15 cm soil depth, K0Na1 reached the soil desalination prerequisite…" -What could be the reason for the increase in bulk electrical conductivity at 15 cm in K0Na1 after 4th irrigation?

**Response:** The decrease in the soil macro-porosity, soil water retention, and weaker hydraulic conductivity all contribute to the increase in bulk electrical conductivity of K0Na1. A greater reduction in $Na^+$ concentration was associated with a higher rate of cation exchange rate, and the slow rate of solute leaching from aggregates reduced the total leaching efficiency (Shaygan et al., 2017). During the leaching process, water flow preferentially passed through the macropores rather than aggregates. The slow water transportation through aggregates induced the slow removal of solutes from the aggregates, leading to a reduced leaching efficiency. In our study, the alternate leaching

was implemented to improve solute leaching. The soil solutes diffused into the aggregates surface during the rest period, improving salt leaching due to the water flow in macropores (Al-Sibai et al., 1997). Increasing the relative ratio of $K^+$ to $Na^+$ could increase the magnitude of cation exchange due to the substitution of $Na^+$ on exchange sites by $K^+$ with lower dispersive potential (Shaygan et al., 2017), the intensive release of cations from the soil further improved salt's leaching efficiency. In addition, the integrity of soil aggregates created by combining clay particles and the other soil components enhanced by $K^+$ can benefit solute transportation (Marchuk and Rengasamy 2011) (Lines 355-369).

**Question 8:** L326: "Therefore, in our study, the high $K^+/Na^+$ ratio promoted the flocculation and stabilization of soil clay particles, resulting in an increased infiltration rate." -What about infiltration rate data? There is no information in the manuscript.

**Response:** This sentence illustrates the data in Fig. 2 (Saturated hydraulic conductivity under different treatments), and to avoid misleading, we have revised this sentence as follows:

Lines 325-327: Therefore, in our study, the high relative concentration of $K^+$ to $Na^+$ promoted the flocculation and stabilization of soil clay particles, resulting in an increased water hydraulic conductivity (Fig. 2).

**Question 9:** I suggest indicating the number of figure/table throughout the discussion (e.g. L332 "The results also implicated…").

**Response:** We have indicated the figure and table number in the discussion.